# META CONTROLNET: ENHANCING TASK ADAPTATION VIA META LEARNING

## ABSTRACT

Diffusion-based image synthesis has attracted extensive attention recently. In particular, ControlNet that uses image-based prompts exhibits powerful capability in image tasks such as canny edge detection and generates images well aligned with these prompts. However, vanilla ControlNet generally requires extensive training of around 5000 steps to achieve a desirable control for a single task. Recent context-learning approaches have improved its adaptability, but mainly for edge-based tasks, and rely on paired examples. Thus, two important open issues are yet to be addressed to reach the full potential of ControlNet: (i) zero-shot control for certain tasks and (ii) faster adaptation for non-edge-based tasks. In this paper, we introduce a novel Meta ControlNet method, which adopts the task-agnostic meta learning technique and features a new layer freezing design. Meta ControlNet significantly reduces learning steps to attain control ability from 5000 to 1000. Further, Meta ControlNet exhibits direct zero-shot adaptability in edge-based tasks without any finetuning, and achieves control within only 100 finetuning steps in more complex non-edge tasks such as Human Pose. [1]

## 1 INTRODUCTION

Image synthesis (Dhariwal & Nichol, 2021; Ramesh et al., 2022; Brock et al., 2018) is a rapidly growing field in computer vision and draws significant interest from various application domains. As a key approach in this area, Generative Adversarial Networks (GANs) (Brock et al., 2018; Karras et al., 2019; Goodfellow et al., 2020) employ a discriminator-generator pair, where the generator is trained to creating enhanced images via sharpening the discriminator. However, such an adversarial approach typically has difficulty to model more complex distributions. Recently, diffusion models (Dhariwal & Nichol, 2021; Ramesh et al., 2022) have emerged as a powerful alternative, excelling in high-quality image generation. These models utilize a series of denoising autoencoders to progressively refine an image from pure Gaussian noise. Among these, a new model known as Stable Diffusion (Rombach et al., 2022) has been proposed, which has better computational efficiency. Unlike traditional methods, Stable Diffusion uses latent representations for image compression, and achieves superior image quality, which includes advancements in text-to-image synthesis and unconditional image generation.

ControlNet (Zhang et al., 2023) further advances image synthesis with enhanced control over image content by using conditional control as different tasks. This approach clones the encoder and middle block of Stable Diffusion, and introduces zero convolution to link with the decoders of Stable Diffusion. Such a setup allows ControlNet to accept image prompt inputs, such as canny or HED edge, and can generate images specific to certain tasks, demonstrating improved control from both image and textual inputs. However, ControlNet's capability for precise control requires extensive training. Specifically, learning to control a new task demands about 5000 steps. Recently, Prompt Diffusion was proposed in (Wang et al., 2023), which leverages in-context learning idea to enhance ControlNet's adaptability to new tasks, but requires task-specific example pairs for training.

Although ControlNet and its variants have achieved enhanced the generalizability, several critical open issues remain unresolved to reach the full power of ControlNet. **Firstly**, *zero-shot* capability of ControlNet has not yet been explored, leaving it as an open question whether it is possible to control new tasks without finetuning samples. **Secondly**, while most existing studies have focused on

---

[1]Our code is provided at `https://anonymous.4open.science/r/meta_controlnet-4E0C`

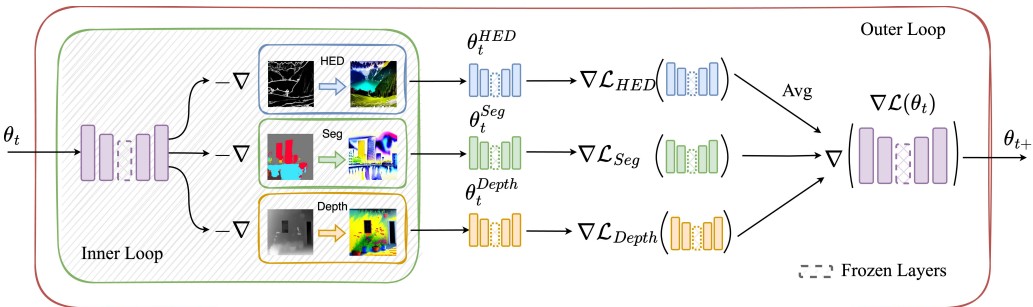

Figure 1: Trained from stable diffusion initial $\theta_{SD}$, the meta learned initial $\theta_{meta}$ is used for various task adaptation.

edge-based tasks, rapid adaptation in more complex scenarios, such as the human pose task, has not yet been achieved.

Figure 2: Meta ControlNet training pipeline. ControlNet parameter is meta updated via meta tasks (HED, Segmentation, Depth). Stable Diffusion parameters are fixed and ControlNet middle layers (Encoder Block 4 and Middle Block) are frozen during the training phase.

In this paper, we propose a novel **Meta ControlNet** method to address the aforementioned open issues. Specifically, Meta ControlNet adopts the FO-MAML (Finn et al., 2017) framework with various image condition types serving as different meta tasks. The inner-loop training of Meta ControlNet takes finetune steps separately for each task. Then the outer-loop training updates meta parameters (i.e., the model initial) based on averaged gradients over all training tasks.

**(Novel Layer Freezing Design)** Meta ControlNet features a new layer freezing design. Typically, meta learning algorithms such as ANIL (Raghu et al., 2019) freezes the earlier embedding layers during the inner-loop training. As a sharp difference, Meta ControlNet freezes *latter* encoder block and the *middle* block during meta training. This idea is based on the observation that the initial encoder blocks are directly linked to the control images of control tasks. It is essential to finetune these encoder blocks for individual task. On the other hand, the middle and latter encoder blocks, which capture common and high-level information, can be retained and shared across tasks. Such a design has been proven to be critical for Meta ControlNet to exhibit desirable performance in our experiments. Note that for the meta testing phase, we recommend training all layers to achieve the best possible adaptation performance.

In the following, we highlight the superior experimental performance that Meta ControlNet achieves:

- **Fast Learning of Control Ability:** Our proposed design significantly enhances the efficiency of ControlNet's learning process. Our experiments demonstrate that Meta ControlNet acquires control abilities within only 1000 steps, a stark improvement over the vanilla ControlNet that achieves the same ability with 5000 steps. Meanwhile, this efficiency is demonstrated across three meta training tasks, showcasing the method's versatility.

- **Zero-Shot Adaptation for Edge-based Tasks:** Meta ControlNet produces generalizable model initial, which exhibits exceptional control capabilities in zero-shot settings, especially for edge-

based tasks such as the canny task. This indicates that our model can adapt to new edge-based tasks without any task-specific finetuning. This is the *first* achievement of successful zero-shot adaptation by ControlNet. Our experiments also indicate that few-shot finetuning often further enhances the fidelity of generated images.

- **Fast Adaptation in Non-edge Tasks:** For challenging tasks in few-shot contexts, our learned model initial exhibits a robust adaptation ability. For instance, it can adapt to the human pose task within merely 100 steps and excel in the more complex human pose mapping task in only 200 steps. These achievements not only surpass all existing benchmarks but also substantially reduce image sample number required for adaptation.

## 2 RELATED WORK

### 2.1 DIFFUSION MODEL AND CONTROLNET

With the recent advancement of score-based generative models (Ho et al., 2020; Song & Ermon, 2019; Song et al., 2020b;a), diffusion models have achieved remarkable performance in text-to-image synthesis (Dhariwal & Nichol, 2021; Ramesh et al., 2022). Essentially, the diffusion model learns a time-varying mapping that gradually transforms a random noise into the sample space via a reverse diffusion process (Ho et al., 2020). Stable Diffusion (SD) (Rombach et al., 2022), as an important step to achieve high-resolution image generation, utilizes a variational autoencoder to first encode images into a latent space, and then learns a time-conditioned U-Net to perform the denoising process on the latent space based on text prompts.

To allow diffusion models to receive more diverse user-specific guidance for image generation, Composer (Huang et al., 2023), ControlNet (Zhang et al., 2023), GLIGEN (Li et al., 2023b) and T2I-Adapter (Mou et al., 2023) were proposed as general approaches to introduce additional controlling signals. Among these, ControlNet (Zhang et al., 2023) stands out by its superior performance in various downstream tasks ranging from sketch (edge, skeleton) to geometry (depth, normal) guided generation. Technically, ControlNet freezes the original SD model, while finetuning a duplicate of the pre-trained SD to integrate additional control signals via zero-initialized convolution modules. Such an adaptation scheme significantly reduces training costs by re-using the image prior learned in the pre-trained SD. ControlNet-XS (Rother, 2023) further investigates the size and architectural design of ControlNet and proposes a more parameter-efficient architecture. As downstream applications, by leveraging ControlNet, Goel et al. (2023) is able to edit the structure and appearance properties for each object in the image, and Chu et al. (2023); Wu et al. (2023) enforces temporal consistency for video generation, Ma et al. (2023); Seo et al. (2023) make pre-trained SD aware of 3D knowledge and multi-view geometry. However, for each of these tasks, an independent adapter is required for each condition. The modified version Multi-ControlNet (Zhang et al., 2023) demonstrates the possibility of composing multiple tasks. Uni-ControlNet (Zhao et al., 2023) proposes a unified framework allowing for the simultaneous utilization of different local and global controls. Prompt Diffusion (Wang et al., 2023) trains an open-domain ControlNet in an in-context learning manner. However, none of these studies are designed for or have been demonstrated to have zero-shot generalization capability for unseen control tasks.

### 2.2 META LEARNING

Meta learning focuses on few-shot learning scenarios, aiming to develop algorithms that leverage a large set of pre-defined tasks to improve performance on unseen instances with only a few or even zero extra training data samples. In this paper, we focus on "learning-to-initialize" approaches such as Model-Agnostic Meta Learning (MAML) (Finn et al., 2017), which learns an initialization point from which models can fast adapt to new tasks. MAML (Finn et al., 2017) algorithm involves two layers of training, where at each iteration, the inner loop optimizes the parameter for each task independently starting from the current initialization, whereas the outer loop estimates the gradient with respect to the inner loop optimization path to update the initialization. Since differentiating through an optimization algorithm is often computationally expensive, FO-MAML (Finn et al., 2017) proposes to simplify the outer-loop gradient computation by directly averaging task-specific gradients evaluated at the outputs of the inner loop. ANIL (Raghu et al., 2019) improves MAML via feature reusing, where the main feature backbone is frozen and only prediction heads are updated by each task in the

inner loop. Reptile (Nichol et al., 2018; Nichol & Schulman, 2018) simplifies the process by aiming for an initialization that minimizes the expected loss across all tasks, similar to joint training.

In computer vision, a popular application of meta learning is few-shot image classification, where a network is adapted to new classes using only a small number of labeled instances per class (Finn et al., 2017; Ravi & Larochelle, 2016; Verma et al., 2020). Li et al. (2018) also adopts MAML-based methods to improve cross-domain generalization. MetaGAN (Zhang et al., 2018) and MetaDiff (Zhang & Yu, 2023) combines MAML respectively with GAN (Goodfellow et al., 2020) and diffusion model (Song & Ermon, 2019; Ho et al., 2020) to facilitate few-shot image classification. More recently, meta learning has been employed to accelerate implicit neural representation for visual signals (Sitzmann et al., 2020; Tancik et al., 2021).

## 3 META CONTROLNET

In this section, we first propose the Meta ControlNet method, and then explain how to select and structure both training and adaptation tasks.

### 3.1 ALGORITHM DESIGN

In this section, we propose our algorithm Meta ControlNet, which maintains the Stable Diffusion network while training its duplicates via the task-agnostic meta learning technique for obtaining adaptive model initial.

In particular, Meta ControlNet adopts three control tasks (HED, Segmentation, Depth) as the primary meta tasks. The training of Meta ControlNet takes the double-loop training framework of FO-MAML (Finn et al., 2017), as depicted in Figure 2, and is described in detail as follows.

The *inner-loop* training of Meta ControlNet takes finetune steps separately for each task. During each step $t$, the meta parameter $\theta_t$ of Meta ControlNet is finetuned independently for each task based on gradient descent as follows:

$$\text{(Inner Loop)} \quad \theta_t^{task} = \theta_t - \alpha \nabla \mathcal{L}_{task}(\theta_t),$$

where $task \in \{\text{HED, Seg, Depth}\}$ and $\alpha$ represents the step size. Note that we here update the parameter only once in the inner loop to enhance efficiency.

The *outer-loop* training first calculates the meta gradient $\nabla \mathcal{L}(\theta_t)$ as follows by taking an average of the gradients across all tasks, based on each task's finetuned parameters in the inner loop:

$$\nabla \mathcal{L}(\theta_t) = \text{Avg}_{task}(\nabla \mathcal{L}_{task}(\theta_t^{task})),$$

where "Avg" denotes the averaging operator over all task gradients. Then the meta parameter $\theta_t$ is updated using the meta gradient as follows:

$$\text{(Outer Loop)} \quad \theta_{t+1} = \theta_t - \alpha \nabla \mathcal{L}(\theta_t),$$

where $\alpha$ is the step size. This design guides Meta ControlNet to minimize the loss of the finetuned model for each task, and hence makes the model more responsive to updates and enables fast adaptability.

**Novel Layer Freezing Design:** A main novel component that Meta ControlNet features is the design of freezing layers during training, which turns out to be critical for its superior performance. Typically, meta learning algorithms such as ANIL (Raghu et al., 2019) freeze the earlier embedding layers during the inner-loop training. Meta ControlNet has sharp differences in two aspects. **Firstly,** Meta ControlNet freezes *latter* encoder block and the *middle* block during meta training. This idea is based on the observation that the initial encoder blocks are directly linked to the control images of control tasks. Given that our Stable Diffusion initial was used to process Gaussian noise rather than control task-specific inputs, and different control image styles signify distinct tasks, it is essential to finetune these initial encoder blocks for individual tasks. On the other hand, the middle and latter encoder blocks, which capture common and high-level information, can be retained and shared across tasks. Therefore, during the training process, Meta ControlNet selectively freezes the Encoder Block4 and Middle block of the U-Net, and focuses on training the remaining parameters. The detailed architecture is available in Appendix. **Secondly,** unlike ANIL, where freezing occurs only in the

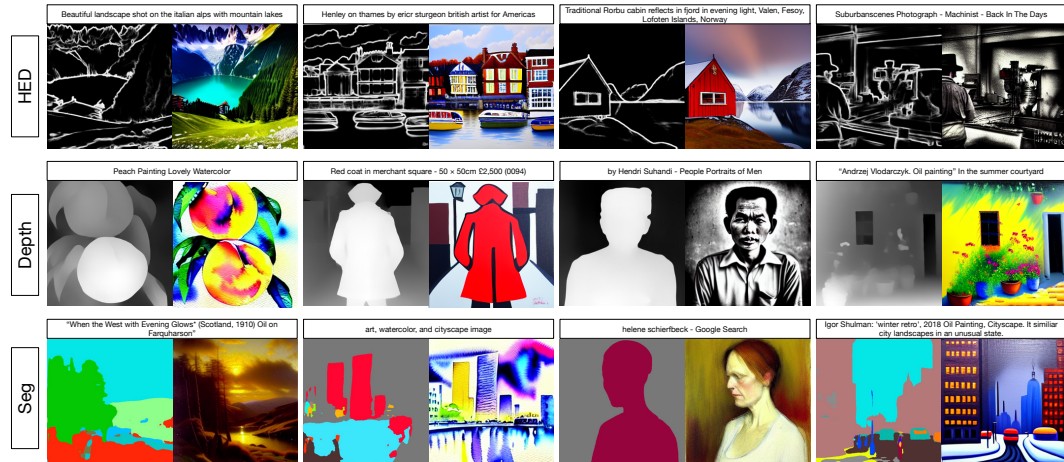

Figure 3: Validation set samples from each training task (HED, Depth, Segmentation) after 1000 steps of training updates.

inner loop, Meta ControlNet applies layer freezing in both inner and outer loops to achieve better efficiency by leveraging our network's initial high-quality image generation capability from Stable Diffusion.

It is important to note that the meta design and layer freezing are applied only during Meta ControlNet's training phase. During the adaptation phase, finetuning uses the standard ControlNet training protocol without using layer freezing or meta learning methods.

## 3.2 TASK SELECTION

**Training Tasks:** During training phase, we choose HED, Segmentation, and Depth map control as our training tasks. Specifically, we obtain HED map using HED boundary detector proposed by (Xie & Tu, 2015). We obtain Segmentation map using Uniformer (Li et al., 2023a). We collect Depth map using Midas (Ranftl et al., 2020).

**Adaptation Tasks:** Follwoing Wang et al. (2023), we utilize Canny Edge maps and Normal maps as our adaptation tasks. These tasks, which align the generated image with the control image's edges, are categorized as *edge-based* tasks. Additionally, we introduce two more complex tasks to demonstrate the versatility of our model: Human Pose (line segments to objects) and Human Pose Mapping (objects to line segments), referred to as *non-edge* tasks.

In detail, we collect Canny Edge by using Canny Edge detector (Canny, 1986). We obtain Normal map by applying Midas (Ranftl et al., 2020). We collect human pose and its reverse human pose mapping by using Openpose (Cao et al., 2019).

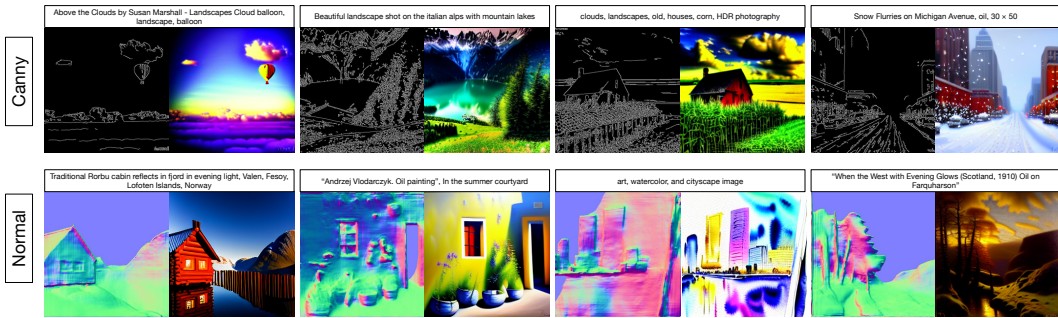

Figure 4: Samples from edge-based tasks (Canny, Normal) in zero-shot adaptation.

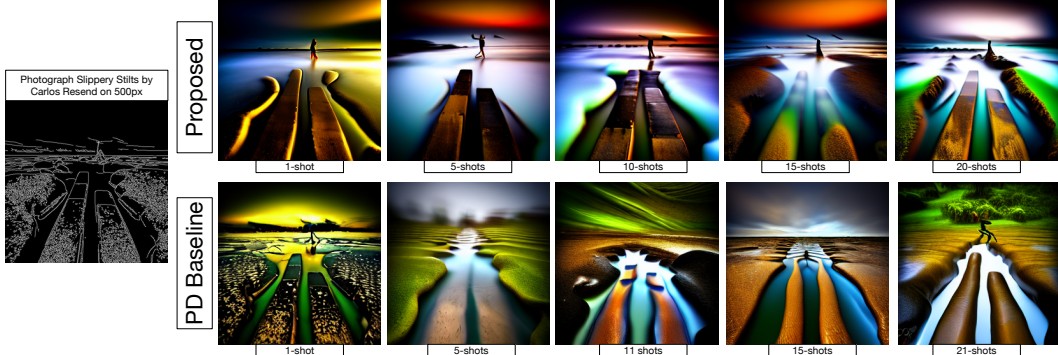

Figure 5: Sample comparison between proposed Meta ControlNet and Prompt Diffusion (PD) baseline for canny task in few-shot finetuning. PD requires example pairs to update and thus is only available in odd number few-shot setting.

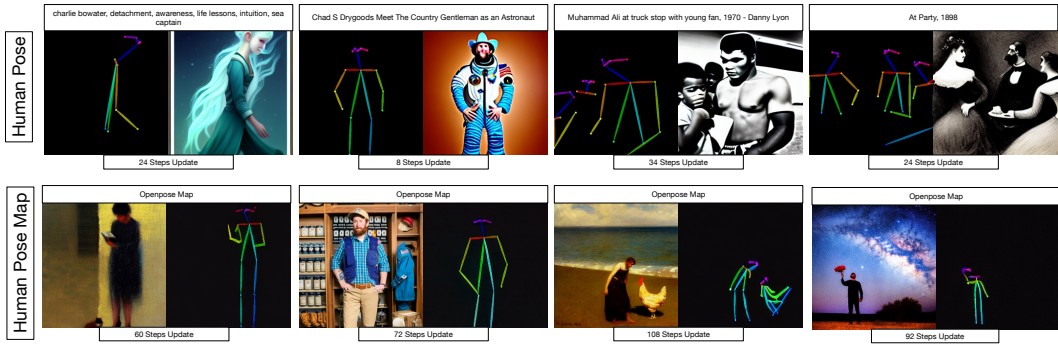

Figure 6: Samples from the validation set for non-edge tasks (Human Pose, Human Pose Map) in a finetuning context. Below each image, the number of updates indicates the first iteration achieving significant control. The validation set is evaluated every two updates.

## 4 EXPERIMENTAL RESULTS

**Dataset:** We use the generated CLIP-filtered dataset proposed by InstructPix2Pix (Brooks et al., 2023) as our training and validation datasets. The CLIP-filtered dataset contains 313k image-prompt pairs.

**Implementations:** Meta ControlNet is developed using the ControlNet codebase (Zhang et al., 2023), and utilizes the Stable Diffusion v1.5 checkpoint for finetuning. We fix learning rate to be $1 \times 10^{-4}$ and batch size to be 256, and accumulate gradients over every 4 batches. Our model is trained on 4 Nvidia A100 GPUs. In our study, both our Meta ControlNet and the baseline of Prompt Diffusion (PD) (Wang et al., 2023) are evaluated at the 8000-step checkpoint. Note that more finetuning steps enhance image quality.

Regarding the meta design, the meta training images in the inner loop are reused in the meta testing phase in the outer loop for each task in order to optimize memory efficiency. Note that the aforementioned batch size of 256 is the total number of images from all tasks. Namely, we randomly sample 256 images across all tasks, and organize them into batches respectively for each task. This strategy ensures that our algorithm does not require additional image sample during the training phase.

### 4.1 FAST CONTROL ACQUIRING IN TRAINING

The proposed Meta ControlNet is trained on tasks of HED, segmentation, and depth mapping. Figure 3 displays the validation results after 1000 steps. Clearly, our Meta ControlNet generates the images that closely match the control images with high fidelity. While the vanilla ControlNet requires

5000 steps to exhibit control ability on a single task, our algorithm Meta ControlNet enjoys rapid learning within only 1000 steps. In fact, for most images, control ability of Meta ControlNet occurs within 500 steps or even fewer, with additional training serving to enhance image fidelity.

## 4.2 ZERO-SHOT CAPABILITY FOR EDGE-BASED TASKS

We evaluate Meta ControlNet on edge-based tasks, specifically Canny and Normal tasks. Figure 4 presents the control images alongside the corresponding generated images and text prompts. This adaptation is assessed in a *zero-shot* context, and does not require finetuning or additional data. This is the *first* ControlNet-type method featuring *zero-shot* capability. In contrast, the baseline Prompt Diffusion (PD) method (Wang et al., 2023) relies on example pairs for learning, rendering it unsuitable for zero-shot settings. The zero-shot results clearly showcase the superior adaptation capability of Meta ControlNet with strong control ability and high fidelity in both tasks.

Further, we compare our Meta ControlNet and the PD baseline with both finetuned in a few-shot context, namely, each method is updated with an equal number of few-shot images. Note that the proposed Meta ControlNet needs only one sample per update step, while PD requires two examples per step. The results in Figure 5 indicate that image quality by Meta ControlNet is enhanced with additional shots. Further, our Meta ControlNet clearly outperforms PD, although the fidelity of the PD gradually improves with more shots. When comparing both methods over the same number of finetuning steps, such as 10 shots for our proposed method and 21 shots for the PD baseline, our approach consistently yields higher quality images. Note that PD requires example pairs to update and thus is only available in odd number few-shot settings.

We highlight that in our experiments, most images generated in zero-shot already exhibit high fidelity, and thus additional few-shot finetuning is not required and might not result in substantial improvements.

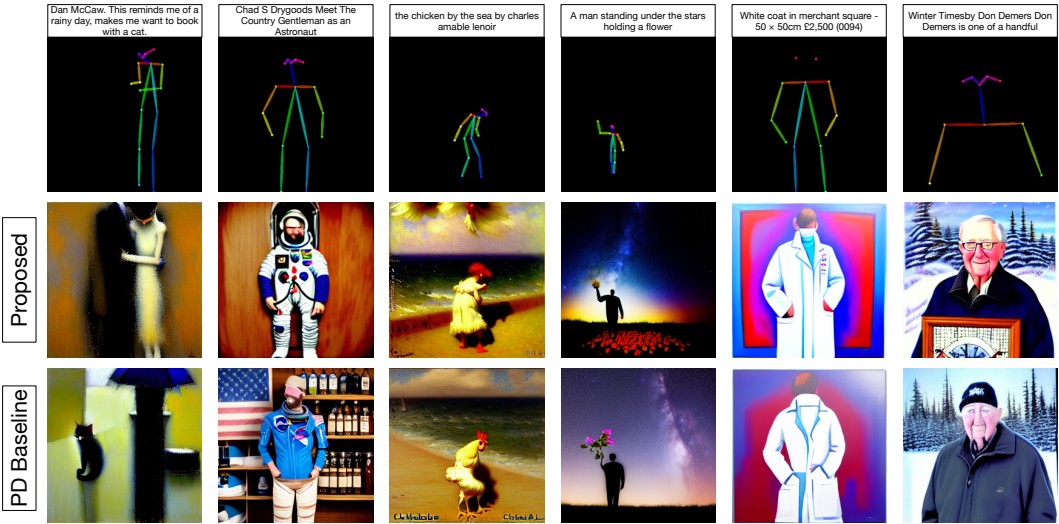

Figure 7: Validation sample comparison between proposed Meta ControlNet and Prompt Diffusion (PD) baseline for Human Pose task after 100 steps of finetuning updates.

## 4.3 FAST ADAPTATION FOR NON-EDGE TASKS

We evaluate the generalizability of Meta ControlNet in non-edge tasks, with focus on human pose and its reverse mapping. These non-edge tasks, which are typically challenging in few-shot setting, require a training-like approach for finetuning. To enhance stability, we double the gradient accumulation from 4 to 8 and keep the batch size to be 256.

For Meta ControlNet, we assess validation at every two steps, and record the first instance when the generated image is aligned with the control image, as shown in Figure 6. We observe that for the human pose task, effective control is achieved within 50 steps. The human pose mapping task,

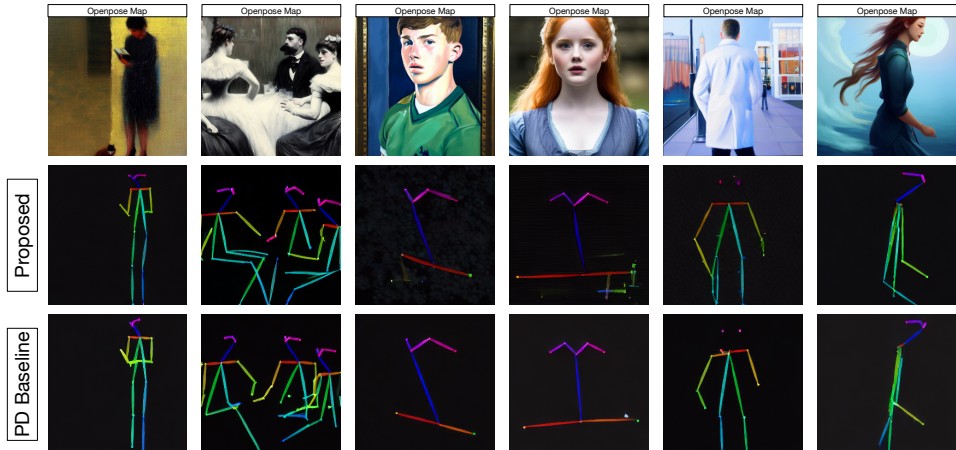

Figure 8: Validation sample comparison between proposed Meta ControlNet and Prompt Diffusion (PD) baseline for Human Pose Mapping task after 200 steps of finetuning updates.

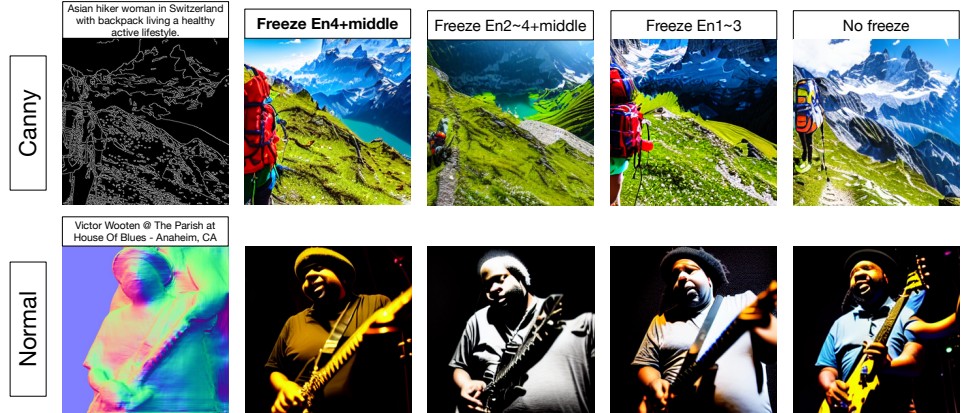

Figure 9: Sample comparison among different freeze methods for edge-based tasks in zero-shot context. $En_N$ refers to the $N^{th}$ Encoder Block. 'Freeze $En_4$ + Middle' refers to freezing the $4^{th}$ Encoder Block and the Middle Block in U-Net, which is adapted in Meta ControlNet.

which converts human poses into line segments, presents a greater challenge due to the deviation from the high-quality images typically generated by stable diffusion. Nevertheless, Meta ControlNet demonstrates control over most samples within approximately 100 steps. We note that in tasks such as human pose mapping, minor errors can occur, for example, incorrectly depicting a chicken pose in the third sample pair. This is due to ControlNet's inherent limitation in distinguishing between human and animal, a distinction that requires learning from more samples.

To compare our method with the PD baseline in the human pose task, we evaluate 100-step finetuning in Figure 7. Despite the use of example pairs, PD achieves control but at the cost of reduced fidelity. In contrast, Meta ControlNet maintains both high control and fidelity. For the more challenging human pose mapping task, our method achieves comparable results as the PD baseline with the same 200 steps, but with only half number of images used by PD, demonstrating our better efficiency.

We note that the different convergence speeds to reach control between the human pose task and its mapping counterpart, i.e., 100 versus 200 steps, arise from the inherent characteristics of Stable Diffusion and our choice of training tasks. Both the standard Stable Diffusion and our selected tasks are geared towards generating natural images rather than line segments. Consequently, in the adaptation phase, the model more readily adapts to the human pose task, which involves creating natural human images, as opposed to learning to generate line segments from human images, a

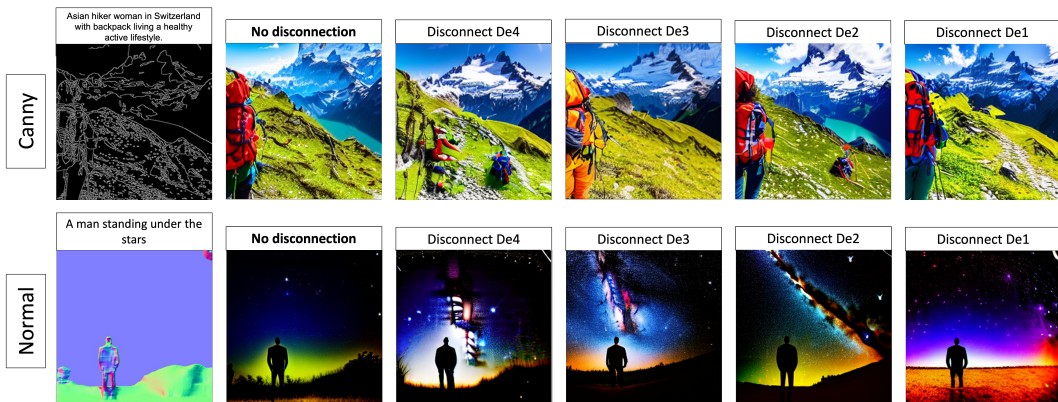

Figure 10: Sample comparison among various connection methods for zero-shot edge-based tasks. $\text{De}_N$ refers to the $N^{\text{th}}$ Decoder Block.

requirement of the mapping task. Nevertheless, our method still achieves better efficiency than PD, namely, comparable results but with only half number of images during training.

## 5    ABLATION STUDY

### 5.1    LAYER FREEZING

To evaluate the effectiveness of various freezing designs, we conduct an experimental comparison focusing on the canny and normal tasks in a zero-shot setting. The experiment compares our freezing strategy (freezing Encoder Block 4 and the Middle block in U-Net) against other methods: freezing Encoder Blocks 2 to 4 plus the Middle block, freezing Encoder Blocks 1 to 3, and no freezing. The U-Net architecture is available in Appendix. The comprehensive results are detailed in Figure 9, with all methods under identical experimental conditions.

Our results indicate that while all freezing designs facilitate control ability, freezing Encoder Block 4 and the Middle block achieves the most accurate alignment with the control image and the highest image fidelity. This superiority is likely because the first three encoder blocks in U-Net are more task-specific, as they are more directly connected to the control image. In contrast, the latter encoder block and the Middle block contribute significantly to the high-quality image output inherent in stable diffusion, making them ideal candidates for freezing across diverse tasks.

### 5.2    DECODER CONNECTION

We evaluate our algorithm using different types of connections between the ControlNet decoder and the pre-trained Stable Diffusion (SD) model decoder. Specifically, we use Meta ControlNet with all decoder connected by zero convolution as our baseline. We then disconnect Decoder Blocks 4 to 1 from the pre-trained SD model in separate variants (each Decoder Block $n$ corresponds to Encoder Block $n$). These variants are trained under the same conditions as the baseline, up to the 8000-step checkpoint, and are then tested on normal and canny edge tasks. Figure 10 depicts the result and suggests that disconnecting Decoder Block 4 significantly reduces image fidelity, and often results in images with repetitive lines or objects with strange shapes. In contrast, disconnecting the other three blocks produces results similar to our baseline. This indicates that Decoder Block 4 plays a critical role in ensuring high image fidelity, which is consistent with our design choice of freezing Encoder Block 4 for this purpose.

## 6    CONCLUSION

In our study, we propose a novel Meta ControlNet approach, which adopts the meta learning technique and features novel freezing layer design to learn a generalizable ControlNet initial. This method

exhibits rapid training convergence, and requires only 1000 steps to effectively control generative imaging. Further, such a meta initial exhibits remarkable zero-shot adaptability for edge-based tasks, the first demonstration in this domain. It also excels in more challenging non-edge tasks, and adapts rapidly within 100 steps for the human pose task and 200 steps for the human pose map task. These results not only outperform existing baselines in terms of control ability and efficiency but also represent significant advancement beyond vanilla ControlNet.

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

## A    APPENDIX

### A.1    DETAILED META CONTROLNET ARCHITECTURE

In Figure 11, the detailed Meta ControlNet architecture is illustrated. This architecture shows that during both training and testing phases, the stable diffusion part remain locked, in line with the ControlNet settings. Specifically, for the ControlNet part, SD Encoder Block4 and SD Middle Block are frozen during the meta training phase, while other blocks are subject to finetuning. This approach is chosen because SD Encoder Blocks 1-3 are more closely linked to the control image, necessitating their adaptation to capture task-specific differences. During the meta testing phase, all blocks within the ControlNet part are finetuned to optimize performance.

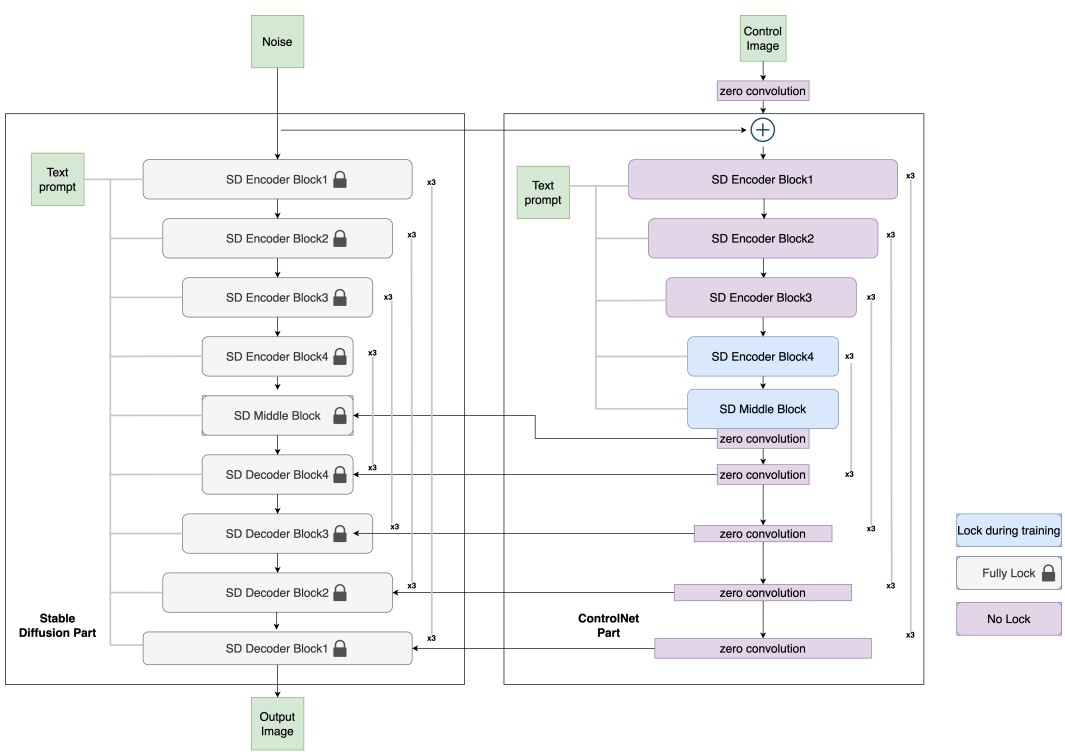

Figure 11: Detailed architecture of Meta ControlNet

### A.2    QUANTITATIVE RESULTS

We quantitatively compare Meta ControlNet with Prompt Diffusion using DreamSim score (Fu et al., 2023), a visual quality metric derived from Large Vision Models such as DINO (Caron et al., 2021), CLIP (Radford et al., 2021), and OpenCLIP (Cherti et al., 2023) that highly aligns with human preferences on visual quality evaluation for both synthetic and real images. We sample the same set of 1130 random images from the test split of InstructPix2Pix (Brooks et al., 2023) for both Prompt Diffusion and Meta ControlNet. Both of the methods also use the same text prompt and the control image for generation. We then measure the average DreamSim score (with different backbone Large Vision Models) over all the generated images and the reference test images for both Meta ControlNet and Prompt Diffusion. We show the Test Average DreamSim Score in Table 1 and Table 2, and Zero-Shot Average DreamSim Score in Table 3 and Table 4. We observe that Meta ControlNet with 1000 training steps achieves significantly better performance than Prompt Diffusion with the same number of training steps over all control tasks from both test generalization and zero-shot generalization perspectives using any DreamSim variations.

| Test Set Generalization | Average DreamSim Score (ensemble) ↓ | | | Average DreamSim Score (DINO-ViTb16) ↓ | | |
|---|---|---|---|---|---|---|
| | HED-to-image | Seg-to-image | Depth-to-image | HED-to-image | Seg-to-image | Depth-to-image |
| Prompt Diffusion (1000 steps) | 0.8182 | 0.8214 | 0.8354 | 0.8389 | 0.8320 | 0.8498 |
| Meta ControlNet (1000 steps) | 0.2905 | 0.3558 | 0.3369 | 0.3225 | 0.3806 | 0.3623 |

Table 1: Test Average DreamSim Scores using Ensemble[2] and Dino-ViTb16 Models as backbone, evaluated under HED, Segmentation, and Depth control tasks.

| Test Set Generalization | Average DreamSim Score (CLIP-ViTb32) ↓ | | | Average DreamSim Score (OpenCLIP-ViTb32) ↓ | | |
|---|---|---|---|---|---|---|
| | HED-to-image | Seg-to-image | Depth-to-image | HED-to-image | Seg-to-image | Depth-to-image |
| Prompt Diffusion (1000 steps) | 0.8338 | 0.8393 | 0.8467 | 0.8296 | 0.8338 | 0.8416 |
| Meta ControlNet (1000 steps) | 0.2573 | 0.3182 | 0.2969 | 0.2615 | 0.3217 | 0.2947 |

Table 2: Test Average DreamSim Scores using CLIP-ViTb32 and OpenCLIP-ViTb32 Models as backbone, evaluated under HED, Segmentation, and Depth control tasks.

| Zero Shot Generalization | Average DreamSim Score (ensemble) ↓ | | Average DreamSim Score (DINO-ViTb16) ↓ | |
|---|---|---|---|---|
| | Canny-to-image | Normal-to-image | Canny-to-image | Normal-to-image |
| Prompt Diffusion (1000 steps) | 0.8179 | 0.8329 | 0.8409 | 0.8580 |
| Meta ControlNet (1000 steps) | 0.3358 | 0.3858 | 0.3636 | 0.4081 |

Table 3: Zero-Shot Average DreamSim Scores using Ensemble and Dino-ViTb16 Models as backbone, evaluated under Canny and Normal map control tasks.

| Zero Shot Generalization | Average DreamSim Score (CLIP-ViTb32) ↓ | | Average DreamSim Score (OpenCLIP-ViTb32) ↓ | |
|---|---|---|---|---|
| | Canny-to-image | Normal-to-image | Canny-to-image | Normal-to-image |
| Prompt Diffusion (1000 steps) | 0.8404 | 0.8459 | 0.8261 | 0.8396 |
| Meta ControlNet (1000 steps) | 0.2975 | 0.3461 | 0.3017 | 0.3531 |

Table 4: Zero-Shot Average DreamSim Scores using CLIP-ViTb32 and OpenCLIP-ViTb32 Models as backbone, evaluated under Canny and Normal map control tasks.

---

[2]By default, DreamSim uses an ensemble of CLIP, DINO, and OpenCLIP (all ViT-B/16) to achieve the best human preference alignment on visual quality evaluation.

