# OpenReview forum: "Meta ControlNet: Enhancing Task Adaptation via Meta Learning"
_ICLR.cc/2025/Conference — ICLR 2025 Conference Withdrawn Submission_

### Official Review · Reviewer_7d2z · 2024-10-29

**Soundness:** 3
**Presentation:** 1
**Contribution:** 2
**Rating:** 5
**Confidence:** 3

**Summary:**

This paper introduces a Meta ControlNet method, which adopts the task-agnostic meta learning technique and features a new layer freezing design. The proposed model significantly enhances the efficiency of ControlNet’s learning process. Meta ControlNet model produces a generalizable model initial, which exhibits exceptional control capabilities in zero-shot settings. The proposed Meta  ControlNet experiment results also achieve comparable performance.

**Strengths:**

The motivation of this paper is very interetsting to use task-agnostic meta learning to explore the full potential of ControlNet.

The paper effectively communicates its main ideas. The explanation of how meta-learning can be incorporated into ControlNet is well-presented and offers code and valuable guidance for practical applications.

**Weaknesses:**

1 Limited technical novelty since theory and approach are straightforward. Moreover, the experimental results are not sufficient. The novelty and effort of this paper cannot reach ICLR standards.

2 This paper does not summarize the differences of current related work and clarify their difference.
3 The authors state: A main novel component that Meta ControlNet features is the design of freezing layers during training. What is the motivation for designing the freezing layer, and what benefits can he bring to this model?

4 Experiment settings should follow the ControlNet. The experimental section doesn't provide any quantitative experimental data. The ablation study is not sufficient.In addition, the proposed method is only verified on one dataset and lacks generalization.

5 Three contributions all state the proposed model can enhance the efficiency of the ControlNet’s learning process. The contribution section seems to be missing an analysis of the methodology.

6 This paper is not well-organized, i.e., the figure does not correspond to the page on which the text is located (e.g., figure 7 is on page 7, but the text is on page 8).

**Questions:**

Nope

---

### Official Review · Reviewer_hkKe · 2024-11-03

**Soundness:** 2
**Presentation:** 3
**Contribution:** 2
**Rating:** 5
**Confidence:** 3

**Summary:**

In this paper, the authors aim to enhance ControlNet from two aspects: i) zero-shot control for certain tasks, and ii) faster adaptation for non-edge-based tasks. Specifically, the authors adopt the task-agnostic meta learning technique (i.e., the FO-MAML framework with various image condition types serving as different meta tasks) and feature a new layer freezing design (i.e., freeze latter encoder block and the middle block during meta learning). Experimental results on both edge-based and non-edge-based tasks demonstrate the superior control ability and efficiency of the proposed method.

**Strengths:**

-	The proposed method is well-motivated. Enhancing the task adaptation ability of ControlNet is indeed an interesting research question to explore. The authors make a good attempt in this direction to learn a more generalizable ControlNet initial via the meta learning technique.
-	The paper is generally well-written and easy to follow.

**Weaknesses:**

-	There are only qualitative results in the paper. The authors should provide quantitative results as well to make the work more convincing.
-	For the non-edge-based tasks, the authors only focus on Human Pose. Since one major aspect of this paper is to enable fast adaptation of non-edge-based tasks, the authors should consider other non-edge-based tasks as well.
-	The authors claim that the proposed method is more efficient with fewer inference steps. However, the proposed method also involves an additional training stage. It would be better if the authors could provide the computational cost comparison with other methods including the training stage.

**Questions:**

I have additional questions:

-	The authors choose three tasks (i.e., HED, Segmentation, Depth) as the meta tasks. Could the authors provide a justification for this? What if other tasks are used during meta training? Have the authors ablated the effect of different meta tasks?

Given the current status of the paper, I am leaning towards borderline reject and hope the authors could address my concerns during the rebuttal.

---

> ### Author Response · Authors · 2024-11-25
> **Response to Reviewer hkKe**
>
> We thank Reviewer hkKe for appreciating our motivation to enhancing task adaptation of ControlNet using Meta learning framework. Regarding your questions, please see our detailed responses below:
>
> 1. **No quantitative results are provided:**: We have conducted quantitative comparisons between Prompt Diffusion [1] and our method, showing that our method can achieves better performance and converge fast. A brief comparison is provided in the Common response, and a more detailed comparision is updated in the Appendix section of the revised version of our paper.
>
> 2. **The choice of training tasks:** Our training task selection of Seg, HED, depth follows the baseline Prompt Diffusion [1] where the authors selected these specific tasks along with their inverse tasks for training. We thank the suggestion from the reviewer and will conduct experiments to explore other training tasks choices in the revision.
>
> 3. **Lack training cost comparisions:** We measured the average wall clock time of training per step for both Prompt Diffusion and our method, under the same training hyperparameters such as batch size, accumulate_grad_batches, and number of GPUs for training. We observe that our method is slightly slower in per step training comparisons. However, our method can significantly reduce the number of training steps to obtain similar or even better performance.
>
> | Method            | Average wall clock time per training step (s) |
> |-------------------|------------------------|
> | Prompt Diffusion       | 5.12          |
> | Meta ControlNet   | 5.49         |
>
> References:
>
> [1] Wang, Zhendong, et al. "In-context learning unlocked for diffusion models." NeurIPS (2023).

---

> ### Comment · Reviewer_hkKe · 2024-11-26
> **Official Comment by Reviewer hkKe**
>
> Thanks for the authors' rebuttal. The rebuttal does not well address my concerns about other non-edge-based tasks, the choice of training tasks, etc. After reading the other reviewers' comments, I agree that the current paper does not meet the acceptance threshold of ICLR and still has a lot of room to improve. Thus, I would like to keep my original score. I suggest the authors incorporate all the comments discussed in the rebuttal to further improve the paper and submit it to another venue.

---

### Official Review · Reviewer_uMVX · 2024-11-04

**Soundness:** 2
**Presentation:** 1
**Contribution:** 2
**Rating:** 5
**Confidence:** 4

**Summary:**

This paper introduces Meta ControlNet, which freezes the latter encoder block and the middle block during meta training of ControlNet. The insight of this paper is that, the initial encoder blocks are directly linked to the control images of each task. And the middle and latter encoder blocks capture common and high-level information thus can be retained and shared across tasks. So simply training the initial encoder blocks can largely reduce training overhead during Meta ControlNet’s training phase.

**Strengths:**

(1) This paper focus on an interesting question, which as great significance to downstreaming research and tasks.

(2) The overall design of the model design is generally make sense.

**Weaknesses:**

(1) This paper proposes an efficient ControlNet training paradigm. However, the experiment part only shows visual comparison, without numerical comparison of training time, quantitative comparison on existing benchmark, or user study. This makes the comparison results unreliable.

(2) I would expect results for adaptation tasks that are more challenging. For example, training on pose and adapt to depth map. Since we know that conditions with similar representation type (such as sketch and canny)  can directly adapt to each other without training.

(3) Providing more results of how the proposed method perform compared to other meta learning methods would be helpful for paper quality.

**Questions:**

See weekness.

---

> ### Author Response · Authors · 2024-11-19
> **Response to Reviewer uMVX**
>
> We are grateful to Reviewer uMVX for recognizing our motivation on the model design. We addressed the concerns as follows:
>
> 1. **No quantitative results are provided**: We conducted quantitative comparisons between Prompt Diffusion [1] and our method, showing that our method can achieves better performance and converge faster. A brief comparison is provided in the Common response, and a more detailed comparision is updated in the Appendix section of the revised version of our paper.
>
> 2. **Adaptation tasks are not challenging enough**: Our training task selection of Seg, HED, depth follows the baseline Prompt Diffusion[1] where the authors selected these specific tasks along with their inverse tasks for training. Additionally, these adaptation tasks are experimentally verified in [1] to be **very challenging** for Stable Diffusion models, where finetuning ControlNet on one task cannot lead to high quality generation on a similar task.
>
> 3. **Lack comparison with other meta learning methods**: Concerning our training strategy, it markedly diverges from the FO-MAML or its layer freezing variant, ANIL [2]. In standard meta-learning, layer freezing typically occurs only during the inner-loop, whereas in our approach, we implement freezing in both inner and outer loops. This decision is informed by our choice of a pre-trained stable diffusion model as the starting point, aiming to maintain static embeddings when training control tasks. Since our training strategy requires less parameters to finetune, the proposed Meta ControlNet greatly enhances the convergence speed, reducing the control ability learning from 5000 step to 1000 steps as validated in our experiments.
>
> References:
>
> [1] Wang, Zhendong, et al. "In-context learning unlocked for diffusion models." NeurIPS (2023).
>
> [2] Raghu, Aniruddh, et al. "Rapid learning or feature reuse? towards understanding the effectiveness of maml." ICLR (2020)

---

### Official Review · Reviewer_jw4S · 2024-11-06

**Soundness:** 3
**Presentation:** 3
**Contribution:** 3
**Rating:** 5
**Confidence:** 3

**Summary:**

This paper introduces Meta ControlNet, leveraging meta-learning and a novel layer-freezing approach to significantly reduce the training steps needed for ControlNet from 5000 to 1000. Additionally, it enables zero-shot control in edge tasks and rapid adaptation in complex tasks like Human Pose with only 100 finetuning steps.

**Strengths:**

1. The idea is simple and easy to understand, that is, using meta-learning to train ControlNet from a set of mixed tasks, and further enhance its adaptation capability in new tasks or zero-shot tasks.
2. The visual results look good and training steps are also reduced.

**Weaknesses:**

There are three critical issues for this paper:
1. No quantitative results are provided. The paper only shows qualitative results, which makes it hard to evaluate the performance of the proposed method.
2. Zero-shot capability. This paper claims that the proposed method can achieve zero-shot control in edge tasks. The model is trained on HED, Segmentation, and Depth map which belong to the edge tasks. It raises a question about whether the model has achieved zero-shot control in edge tasks. If we train ControlNet on three tasks, and then test the model on Canny and Normal, can ControlNet achieve similar 'zero-shot' control in these two tasks?
3. Training costs. If there are greater costs in training and we still need to fine-tune the model on new tasks like pose even with fewer steps (the total costs mighit be increased), we need to elborate more on the training costs and its significance.

**Questions:**

NA

---

> ### Author Response · Authors · 2024-11-19
> **Response to Reviewer jw4S**
>
> We thank Reviewer jw4S for the time and effort in reviewing our paper. Per your questions, please see our response below:
>
> 1. **No quantitative results are provided:** We conducted quantitative comparisons between Prompt Diffusion [1] and our method, showing that our method can achieves better performance and converge fast. A brief comparison is provided in the Common response, and a more detailed comparision is updated in the Appendix section of the revised version of our paper.
>
> 2. **Lack of Zero-shot capability:** In our quantitave results, we trained both Prompt Diffusion baseline and our method on three edge-based tasks (HED, Segmentation, and Depth map), and test the model’s zero shot capability on Canny-to-image and Normal-to-image tasks (as well as the test set results from the three training tasks). Our results show that the trained model can do zero-shot generalization on these two unseen tasks.
>
> 3. **Unknown training cost:**
> We measured the average wall clock time of training per step for both Prompt Diffusion and our method, under the same training hyperparameters such as batch size, accumulate_grad_batches, and number of GPUs for training. We observe that our method is slightly slower in per step training comparisons. However, our method can significantly reduce the number of training steps from 5000 to 1000 to obtain similar or even better performance, meaning the whole training cost is less than baseline's counterpart.
>
> | Method            | Average wall clock time per training step (s) |
> |-------------------|------------------------|
> | Prompt Diffusion       | 5.12          |
> | Meta ControlNet   | 5.49         |
>
> References:
>
> [1] Wang, Zhendong, et al. "In-context learning unlocked for diffusion models." NeurIPS (2023).

---

### Author Response · Authors · 2024-11-17
**Common response by Authors**

We thank the reviewers and program chairs for their time and feedback.
We appreciate the reviewers' recognition of our paper's key strengths:
- The strong motivation for using meta-learning to enhance ControlNet's adaptation capabilities (Reviewer **7d2z**, Reviewer **hkKe**,Reviewer **uMVX**).
- The clarity and accessibility of our proposed approach (Reviewer **jw4S**, Reviewer **hkKe**, Reviewer **7d2z**).

In response to the common concern regarding quantitative results: We appreciate all reviewers’ suggestions on including the quantitative results for our method. Therefore, we quantitatively compare Meta-ControlNet with Prompt Diffusion using DreamSim score [1], a visual quality metric derived from Large Vision Models such as DINO, CLIP, and OpenCLIP that highly aligns with human preferences on visual quality evaluation for both synthetic and real images. We sample the same set of 1130 random images from the test split of InstructPix2Pix for both Prompt Diffusion and Meta-ControlNet. Both of the methods also use the same text prompt and the control image for generation. We then measure the average DreamSim score (with different backbone Large Vision Models) over all the generated images and the reference test images for both Meta ControlNet and Prompt Diffusion. We observe that Meta-ControlNet achieves better performance than Prompt Diffusion with the same number of training steps over all control tasks from both test generalization and zero-shot generalization perspectives. We provide two tables for the comparison results as shown below due to character constraints, and we have revised our paper with the full results included in the Appendix section **A.1**.

| Test Set Generalization          | Average DreamSim Score (CLIP-ViTb32) ↓ |                                |                                | Average DreamSim Score (OpenCLIP-ViTb32) ↓ |                                |                                |
|----------------------------------|----------------------------------------|--------------------------------|--------------------------------|------------------------------------------|--------------------------------|--------------------------------|
|                                  | HED-to-image                           | Seg-to-image                   | Depth-to-image                 | HED-to-image                             | Seg-to-image                   | Depth-to-image                 |
| Prompt Diffusion (1000 steps)    | 0.8338                                 | 0.8393                         | 0.8467                         | 0.8296                                   | 0.8338                         | 0.8416                         |
| Meta ControlNet (1000 steps)     | 0.2573                                 | 0.3182                         | 0.2969                         | 0.2615                                   | 0.3217                         | 0.2947                         |


| Zero Shot Generalization         | Average DreamSim Score (ensemble) ↓   |                                | Average DreamSim Score (DINO-ViTb16) ↓ |                                |
|----------------------------------|--------------------------------------|--------------------------------|----------------------------------------|--------------------------------|
|                                  | Canny-to-image                       | Normal-to-image               | Canny-to-image                         | Normal-to-image               |
| Prompt Diffusion (1000 steps)    | 0.8179                               | 0.8329                         | 0.8409                                 | 0.8580                         |
| Meta ControlNet (1000 steps)     | 0.3358                               | 0.3858                         | 0.3636                                 | 0.4081                         |


We will address individual reviewer responses in separate sections.

**References**

[1] Fu, Stephanie, et al. "Dreamsim: Learning new dimensions of human visual similarity using synthetic data." NeurIPS (2023).

---

### Note · Authors · 2024-12-01

I have read and agree with the venue's withdrawal policy on behalf of myself and my co-authors.